# Comparative Analysis of Bioenergy Crop Impacts on Water Quality Using Static and Dynamic Land Use Change Modeling Approach

**Eeshan Kumar †, Dharmendra Saraswat ‡,\* and Gurdeep Singh §**

Department of Biological & Agricultural Engineering, University of Arkansas, Fayetteville, AR 72701, USA; eeshan.kumar@woodplc.com (E.K.); gurdeep.singh@climate.com (G.S.)

\* Correspondence: saraswat@purdue.edu; Tel.: +1-765-494-5013

† Current affiliation: Wood Environment & Infrastructure Solutions, Inc., Wichita, KS 67202, USA.

‡ Current affiliation: Department of Agricultural & Biological Engineering, Purdue University, West Lafayette, IN 47907, USA.

§ Current affiliation: The Climate Corporation, St. Louis, MO 63141, USA.

**Abstract:** Researchers and federal and state agency officials have long been interested in evaluating location-specific impact of bioenergy energy crops on water quality for developing policy interventions. This modeling study examines long-term impact of giant miscanthus and switchgrass on water quality in the Cache River Watershed (CRW) in Arkansas, United States. The bioenergy crops were simulated on marginal lands using two variants of a Soil and Watershed Assessment Tool (SWAT) model. The first SWAT variant was developed using a static (single) land-use layer (regular-SWAT) and for the second, a dynamic land-use change feature was used with multiple land use layers (location-SWAT). Results indicated that the regular-SWAT predicted larger losses for sediment, total phosphorus and total nitrogen when compared to location-SWAT at the watershed outlet. The lower predicted losses from location-SWAT were attributed to its ability to vary marginal land area between 3% and 11% during the 20-year modeling period as opposed to the regular-SWAT that used a fixed percentage of marginal land area (8%) throughout the same period. Overall, this study demonstrates that environmental impacts of bioenergy crops were better assessed using the dynamic land-use representation approach, which would eliminate any unintended prediction bias in the model due to the use of a single land use layer.

**Keywords:** water quality; bioenergy crops; SWAT; marginal lands; dynamic land use change; watershed modeling

## 1. Introduction

The United States Energy Independence and Security Act (EISA) of 2007 was intended to bring energy security through increased biofuel production. Section 204 of the act required the Environment Protection Agency (EPA) to assess impact of biofuel production on local biodiversity, perhaps by soliciting "the views of the National Academy of Sciences or another appropriate independent research institute" [1]. In the meantime, the Biomass Crop Assistance Program (BCAP) was created in the 2008 Farm Bill, to boost production of biofuel crops by providing financial incentives to farmers in 12 states in the US, including Arkansas [2].

A critical driver of interest in biofuel crop production was a target of producing 36 billion gallons of renewable fuels by 2022 [3]. This target was first revised downward to 16.3 billion gallons in 2015 and then increased to 17.4 billion gallons by 2016, respectively, because of a variety of reasons such as constraints in accommodation of increasing volumes of ethanol in the transport fuel market, limitations in the ability of biofuel industries to produce qualifying renewable fuel, etc. [4]. Despite

downward revision of the targeted renewable fuel production from 36 to 17.4 billion gallons by 2022, the need to study environmental footprint of biofuel crop production is still relevant for EPA.

As per the United States Department of Agriculture (USDA), 10.93 million ha (Mha) of cropland would be required to meet the goals of EISA bio-feedstock production [5]. Smeets and Faaij [6] had predicted that by the year 2050, 54 Mha to 348 Mha of surplus agricultural land could be available for bioenergy production. Currently, a major portion of biofuel in the form of ethanol comes from food crops such as (corn, soybean, etc.), which can lead to competition between food and fuel [7] resulting in price increases of agricultural commodities by 26% for cereals, 18% for other crops, and 5% for livestock by 2020 [8]. To decrease this competition, EISA estimated that 15 billion gallons of ethanol from the original target of 36 billion gallons could come from first-generation crops such as sugar crops, starch crops, oilseed crops and animal fats [9]. The remaining 21 billion gallons was expected to be contributed by second-generation biofuel crops comprising cellulosic crops or non-food crops and third generation biofuel sources such as algal biomass [10]. The surplus agricultural land identified by Smeets and Faaij [6] provided motivation for exploring the potential of biofuel crop production. As a result, this surplus land also known as targeted or marginal land has gained attention for bioenergy research [11]. These lands are generally considered marginal for conventional crop production but could be suitable for bioenergy crop production or other functions based on economic, soil health and environmental criteria [12–14]. Cultivation of second-generation biofuel crops on abandoned, degraded, or marginal land has also been reported to decrease competition of cropland for growing bioenergy crops [15].

To assess the environmental impact of bioenergy crops, use of computer-based watershed modeling tools has been gaining momentum since underlying hydrologic and nutrient transport processes are well represented in these models. This research article examines the use of Soil and Water Assessment Tool (SWAT) model for assessing water quality impacts of producing two different bioenergy crops on marginal lands.

Several SWAT model [16] studies have reported the impact of biofuel crop production on water quality. Ng et al. [17] reported that 30% nitrate load decreased by converting half of the Salt Creek Watershed in Illinois to miscanthus compared to corn and soybean and applying nitrogen fertilizer in corn at a rate of 90 kg-N/ha. Wu et al. [18] addressed the relationship between biofuel production and environmental costs by determining the relationship of biomass production and resultant nitrogen loads against switchgrass planting acreage and locations in the James River Basin in Midwestern US. In the Iowa River Basin, Wu and Liu [19] reported that the production of switchgrass or miscanthus resulted in reduced sediment loss compared to corn production. They further reported that the change of land cover from native grass to biofuel crops decreased water yield but increased nitrate-nitrogen load in water bodies. Kim et al. [20] reported that land use change to miscanthus and switchgrass coupled with climate change altered the hydrometeorology of the Yazoo River Basin, Mississippi. A conversion of marginal lands to miscanthus resulted in reduction of sediment, nitrogen, and phosphorus loadings at the watershed outlet in Michigan [21]. All the foregone studies have reported some water quality benefits with the implementation of biofuel crops.

The dynamic land use change feature was introduced in the SWAT2009 version of the SWAT model. A limited number of peer-reviewed manuscripts are available that have used dynamic land-use change features for setting up a SWAT model at a scale of a single [22] to multiple sub-watersheds [23]. Researchers have also been interested in examining the impact of dynamic land use change on NPS pollution in recent years [24,25]. Changing land uses has been reported to alter the hydrology and sedimentation in a watershed [26]. The present study expects to be a good contributor to the existing SWAT literature base on bioenergy crop production by providing a comparative performance assessment of SWAT model developed using static land use (hereafter referred to as regular-SWAT) and dynamic land-use (hereafter referred to as location-SWAT), respectively.

The specific objectives of this study were (1) to develop two variants of SWAT model, regular-SWAT using a single land use layer and location-SWAT using multiple land use layers and (2) comparatively analyze the water quality impacts by simulating biofuel crops on marginal lands at the watershed outlet.

## 2. Materials and Methods

### 2.1. Study Watershed Description

This study was conducted in the Cache River Watershed (CRW) which lies within the White River Basin and located in northeast Arkansas. It is designated by the hydrologic unit code (HUC) 08020302 (Figure 1).

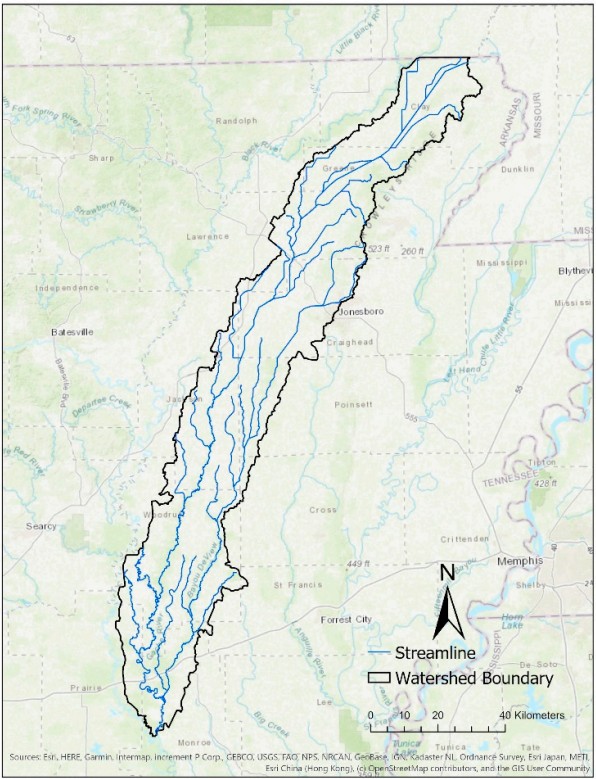

**Figure 1.** Cache River Watershed.

The drainage area of the watershed is 5066 square kilometers. The watershed is about 230 km in length and 29 km at the widest point. About 4403 square kilometers of the watershed lies in the Western Lowlands geological division of the Mississippi Alluvial Valley (MAV), and the remaining portion of the watershed (about 673 square kilometers) lies in the headwater areas along the western slope of Crowley's Ridge. The elevation of the watershed ranges from 44 m above mean sea level at its lowest point to 170 m at its highest point. The CRW is relatively flat with 48 percent of the watershed having slopes ranging from 0 to 1 percent. Major land use and land cover (LULC) in 2006 consisted of soybean (29%), forest (25%), rice (14%), corn (9%), cotton (3%), pasture (3.5%), urban (2.8%), and water (1.6%) [27]. The dominating soil types in the watershed fall in hydrological soil groups C and D (high runoff potential) covering approximately 64% of the watershed. The major soils were Grubbs (12.9%), Calhoun (11.6%), Forestdale (9.6%) and Askew (9.1%). Askew is categorized into hydrologic soils group C, and all other three major soil types are categorized into hydrologic souls group D having high runoff potential.

### 2.2. SWAT Model Description

Computer-based models are effective tools to quantify impacts of feedstock production on hydrology and water quality [28]. They represent 'what-if' scenarios to address questions of long-term impacts relating to land management changes, and their use may be helpful to understand non-

point source levels before any field monitoring [29,30]. SWAT model is one such tool that has been extensively used to predict the effects of different land management scenarios on water quality, pollutant loadings and sediment yields [31–34]. There are more than 3800 peer-reviewed journal articles published with the application of SWAT model since 1984 (SWAT literature database, available at https://www.card.iastate.edu/swat_articles/). Thus, being a well-documented model with a large user base, SWAT was chosen for this study. Borah and Bera [35] had reviewed several continuous simulation models for their ability to model nutrient export with different land management scenarios at a watershed scale and suggested that SWAT was superior to other evaluated models for analyzing long-term impacts of management scenarios and prediction of nutrient loads. Many studies have also provided modeling protocols to evaluate, interpret, and communicate performance of hydrologic/water quality models (including SWAT) considering their intended use [36–38].

SWAT model requires a wide range of inputs that include spatial data (watershed boundary, topography, land use and land cover, soils and stream network), weather data, point source/water quality data, and crop management data. In SWAT, a basin is delineated to subbasins and then to Hydrologic Response Units (HRUs) which represent unique and homogeneous combinations of land use, soil and slope classes within a subbasin and represent percentages of the subbasin area. The HRU delineation for CRW resulted in 14,053 and 12,321 HRUs for the regular-SWAT and the location-SWAT model, respectively. The difference was mainly due to different land use data layers used for setting up the model. Less number of HRUs for location-SWAT compared to regular-SWAT suggests incorporation of unintended bias in the model right from the beginning of simulation. Flows, sediment yield, and nutrient loads that are obtained for each HRU, summed up for each subbasin, are then routed to the outlet of the watershed. Surface runoff is estimated using modified SCS curve number method; sediment yield is estimated with the modified soil loss equation (MUSLE), and nutrient cycles are determined from the EPIC model [39]. A detailed and complete description of all processes simulated in the model and associated required input data are provided in the SWAT theoretical documentation and users' manual [39].

The regular-SWAT model was set up following a traditional practice, i.e., using a single land use layer whereas the location-SWAT model was built using multiple land use layers. The regular-SWAT model used the 2006 land use layer since it was the most recent and most detailed data layer available from the Center for Advanced Spatial Technologies (CAST), University of Arkansas. The other layers used in the location-SWAT model were 1992, 2001, and 2011 layers (available from the National Land Cover Dataset (NLCD)) and 1999 and 2004 layers (available from the CAST). Multiple land use layers were incorporated in the location-SWAT model by activating land use change (LUC) module/land use update (lup.dat) using the SWAT LUC tool (available at https://saraswat-swat.rcac.purdue.edu/) [23]. The LUC module was used to update HRU areas defined by the variable HRU_FR. The value of HRU_FR ranges from 0 to 1 and specifies fractional area occupied by an HRU in a subwatershed.

To simulate and determine water quality impacts represented through Total Phosphorus (TP) and Total Nitrogen (TN) loadings of bioenergy crops, marginal croplands in the watershed were identified. Accurate identification of marginal lands is important to correctly simulate bioenergy crops in a model [40]. In this study, marginal lands were identified based on two criteria: soil health issues (poorly drained and frequently flooded) and land capability classes to generate modified land use layers following an approach developed by Singh and Saraswat [41].

*2.3. Model Sensitivity Analysis, Calibration, Validation, and Uncertainty Analysis*

As SWAT contains several parameters that can be changed, it was important to determine those parameters that greatly affected the model outputs. Thus, a sensitivity analysis was performed to reduce the number of parameters to be adjusted during the calibration of the two models. In this study, a global sensitivity analysis was performed with the SWAT Calibration and Uncertainty Program (SWAT-CUP) software coupled with Sequential Uncertainty Fitting Version 2 (SUFI2) algorithm using a parallel framework on AHPCC (Arkansas High Performance Computing Center) [42]. The advantage of using SUFI2 is that it could be run with a smaller number of model runs, an

important characteristic for computationally demanding models [43]. Twenty-six parameters were selected based on literature review and physical characteristics of the watershed. Nash Sutcliffe Efficiency (NSE) was used as the objective function and 500 simulations were done on the AHPCC.

After identification of sensitive parameters responsible for reducing prediction uncertainty, both models were calibrated to a given set of local conditions by changing selected parameters and comparing the simulated outputs to their measured counterparts [44]. Calibration was followed by validation to demonstrate the models could make sufficient, accurate, site-specific predictions.

In this study, the SWAT models were run for a period of 21 years from 1992 to 2012. A four-year warm-up period was used for both of the models from 1992 to 1995 which is, recommended to initialize and aid in the development of model variables [45]. Considering the availability of observed data (Table 1), the models were calibrated from 1996 to 2005 and validated from 2006 to 2012.

**Table 1.** List of available data for calibration and validation.

| USGS Gage | Drainage Area | Phase | Streamflow | TN | TP |
|---|---|---|---|---|---|
| 07077380 | 1816 km$^2$ | Calibration | 1 January 1996– 31 December 2005 | | |
| | | Validation | 1 January 2006– 31 December 2014 | | 1 January 2013– 31 December 2014 |
| 07077500 | 2694 km$^2$ | Calibration | 1 January 1996– 31 October 1997; 1 October 2002– 31 December 2005 | 1 January 1997– 31 October 1997; 1 October 2002– 31 December 2005 | 1 January 1996– 31 October 1997; 1 October 2002– 31 December 2005 |
| | | Validation | 1 January 2006– 28 February 2011 | 1 January 2006– 28 February 2011 | 1 January 2006– 28 February 2011 |
| 07077555 | 3030 km$^2$ | Calibration | 1 January 1996– 31 December 2005 | | |
| | | Validation | 1 January 2006– 31 December 2014 | | 1 January 2013– 31 December 2014 |

However, there were some periods of missing data within the calibration and validation periods that could impact calibration or validation in unexpected ways. Calibration was first performed on an annual scale to minimize relative error followed by a monthly time scale to account for seasonal trends or variations [22]. Both models were first calibrated at the upstream gauge in the watershed (Egypt) followed by downstream gauges (Patterson and Cotton Plant) to reduce the spatial accumulation of error [43]. Calibration of variables (outputs) was also done in a logical order as recommended by Arnold et al. [44]: hydrologic outputs (total flow, surface runoff, and baseflow) calibrated first because of their influence on other output variables, followed by sediments, total phosphorus (TP), and total nitrogen (TN) [26].

To assess model performance on an annual time scale, relative error (RE) statistic [34] was calculated using the following Equation (1):

$$RE\ (\%) = (|O - P|/O) \times 100 \tag{1}$$

In the above Equation (1), O represents the average annual measured value and P represents the average annual simulated value. To minimize the relative error between observed and simulated values, performance ratings described by Santhi et al. [34] were used to judge annual time scale performance of the models (RE < 15% for average annual measured total flow and RE < 25% for nutrients). Statistical functions such as the coefficient of determination ($R^2$), Nash–Sutcliffe efficiency (NSE), percent bias (PBIAS), and RMSE- observation standard deviation ratio (RSR) were used to evaluate performance of the models on a monthly time scale as recommended by Moriasi et al. [46] and shown in Table 2.

**Table 2.** Performance ratings used for evaluating monthly model results.

| Rating | NSE | RSR | PBIAS (%) | | |
|--------|-----|-----|-----------|---|---|
| | | | Streamflow | Sediment | N, P |
| Very good | $0.75 \leq E \leq 1.00$ | $0.00 \leq RSR \leq 0.50$ | PBIAS ≤ ±10 | PBIAS ≤ ±15 | PBIAS ≤ ±25 |
| Good | $0.65 \leq E \leq 0.75$ | $0.50 \leq RSR \leq 0.60$ | ±10 ≤ PBIAS ≤ ±15 | ±15 ≤ PBIAS ≤ ±30 | ±25 ≤ PBIAS ≤ ±40 |
| Satisfactory | $0.50 \leq E \leq 0.65$ | $0.60 \leq RSR \leq 0.70$ | ±10 ≤ PBIAS ≤ ±25 | ±30 ≤ PBIAS ≤ ±55 | ±40 ≤ PBIAS ≤ ±70 |
| Unsatisfactory | $E < 0.50$ | $RSR < 0.70$ | PBIAS > ±25 | PBIAS > ±55 | PBIAS > ±70 |

Several parameters were used within their recommended ranges (Table 3) to calibrate the model for hydrology.

**Table 3.** Parameters adjusted during calibration.

| Parameter | Description | Unit | Range | Default Value |
|-----------|-------------|------|-------|---------------|
| *Parameters affecting surface water* | | | | |
| CN2 | SCS runoff curve number | none | 35–98 | Varies |
| ESCO | Soil evaporation compensation factor | none | 0–1 | 0.95 |
| CANMX | Canopy storage capacity | mm | 0–100 | 0 |
| *Parameters affecting subsurface water* | | | | |
| ALPHA_BF | Baseflow recession constant | 1/Day | 0–1 | 0.048 |
| GW_REVAP | Ground water revap coefficient | none | 0.02–0.2 | 0.02 |
| GW_DELAY | Ground water delay time | days | 0–500 | 31 |
| REVAPMN | Threshold depth of water in shallow aquifer for percolation | mm | 0–1000 | 750 |
| GWQMN | Threshold depth of water in shallow aquifer for return flow | mm | 0–5000 | 1000 |
| RCHRG_DP | Deep aquifer percolation fraction | none | 0–1 | 0.05 |
| *Parameters affecting phosphorus* | | | | |
| PSP | Phosphorus sorption coefficient | none | 0.01–0.7 | 0.4 |
| SOL_SOLP | Initial soluble P concentration | mg/kg | 0–100 | 5 |
| PHOSKD | Phosphorus soil partitioning coefficient | $m^3$/mg | 100–200 | 175 |
| BC4 | Rate constant for mineralization of organic P | 1/Day | 0.01–0.7 | 0.35 |
| RS5 | Organic P settling rate | 1/Day | 0.001–0.1 | 0.05 |
| BIOMIX | Biological mixing efficiency | none | 0–1 | 0.2 |
| USLE_P | USLE crop practice factor | none | 0–1 | 1 |
| *Parameters affecting nitrogen* | | | | |
| RCN | Concentration of N in rainfall | mg/L | 0–15 | 1 |
| SHALLST_N | Initial concentration of nitrate in shallow aquifer | mg/L | 0–1000 | 0 |
| ERORGN | Organic N enrichment ratio for loading with sediment | none | 0–5 | 0 |
| SDNCO | Denitrification threshold water content | none | 0–1 | 0.8 |
| CDN | Denitrification exponential coefficient | none | 0–3 | 1.4 |
| N_UPDIS | Nitrogen uptake distribution parameter | none | 0–100 | 20 |

After satisfactory calibration for hydrology, both the models were calibrated for TP and TN. Before performing calibration for TP and TN, phosphorus and nitrogen pools (cycles) were initiated using soil test data available for the watershed. In soil, phosphorus (P) is present in three forms: organic phosphorus, mineral phosphorus and plant-available phosphorus. Addition of phosphorus can be done by fertilizer application, manure or residue application. Removal of phosphorus from soil can happen by plant uptake and erosion. In SWAT, soil phosphorus is divided into six pools and is modeled as P mineralization, decomposition, immobilization, sorption of inorganic P, leaching and phosphorus in the shallow aquifer [39]. The SWAT model sets an initial concentration of solution phosphorus (SOL_P) in all layers to 5 mg P kg⁻¹ and 25 mg P kg⁻¹ for unmanaged land under native vegetation and soil for cropland conditions [47]. This initial amount of solution or labile P can be specified by a user. Soil test data were used to initialize phosphorus pools and facilitate phosphorus modeling in SWAT. This data was available for three counties from 1992 to 2012. Percentage of clay and organic carbon were found from the SSURGO database for the top layer of the three major soils.

The following Equation (2) given by Sharpley et al. [48] and Vadas and White [49] was used to calculate phosphorus sorption coefficient (PSP):

$$PSP = -0.053 \times \ln(\% \text{ Clay}) + 0.001 \times (\text{Sol P, mg/kg}) - 0.029 \times (\% \text{ Org Carbon}) + 0.42, \qquad (2)$$

The value of solution P (32.1 mg/kg) was taken as half of the Mehlich 3 test phosphorus values [49]. The final value of PSP and solution P used in the models were 0.28 and 32.1 mg/kg.

Similar to phosphorus, a user can specify the amount of nitrate and organic nitrogen contained in soil layers according to soil test data. Nitrogen is available as organic nitrogen, mineral nitrogen and mineral nitrogen in solution in soils. Like phosphorus, nitrogen can be added by fertilizer application, manure or residue application. Five pools of nitrogen are simulated in SWAT; of these, two are inorganic nitrogen ($NH_4^+$ and $NO_3^-$) pools and the other three are organic forms of nitrogen. SWAT can automatically initialize levels of nitrogen in the different pools if no values of nitrate and organic nitrogen are supplied. Two parameters, namely, concentration of nitrogen in rainfall and initial concentration of nitrate in shallow aquifer were varied based on data available in the watershed. With an intent to provide accurate parameters to the models for better representation of nitrogen levels, these values were changed as described below.

The default value of nitrogen concentration in rainfall (RCN, mg N/L) is set to 1.0 in the model. It was revised to 0.3 mg-N/L for CRW by retrieving atmospheric deposition of nitrogen value from the National Atmospheric Deposition Program (NADP) website (http://nadp.slh.wisc.edu/ntn/annualmapsByYear.aspx#2012). Moreover, the initial concentration of nitrate in shallow aquifer (SHALLST_N, mg N/L) was set equal to 0. Groundwater nitrate concentration value for CRW, available at the county-level was also incorporated in both the models.

After both models were calibrated and validated, an uncertainty analysis was performed using SWAT-CUP with SUFI2. The calibration and uncertainty analysis are combined in SUFI2 to find uncertainty parameters that affect forecast for most of the observed data while providing least possible uncertainty of forecast. It is flexible by allowing arbitrary likelihood/objective functions and could be run with the smallest number of model runs to achieve good prediction uncertainty bands with identified critical sources of uncertainty [43].

AHPCC supercomputer was used to make a total of 1000 simulations performed in two successive iterations of 500 each for both models. After completion of model simulations, SWAT CUP produced results in the form of 95 Predictive Probability Uncertainty (PPU) plots. The degree to which all uncertainties are accounted for is quantified by two measures: P-factor, percentage of measured data bracketed by the 95% prediction uncertainty (95PPU), and r-factor, thickness of the 95PPU band. The goodness of calibration and prediction uncertainty is judged based on the closeness of the P-factor to 100% (indicating observations bracketed by the prediction uncertainty) and the R-factor to 1 (indicating achievement of rather small uncertainty band) [42].

*2.4. Simulation of Bioenergy Crops in SWAT*

Plant growth and development, biomass, yield, nutrient, and water uptake are driven by parameters present in SWAT crop database. In case of perennial plants like switchgrass and miscanthus, crop growth starts when mean daily temperature reaches a base threshold temperature. Moreover, the perennial plants/grasses can maintain a nutrient pool as they do not require replanting and can yield viable crops for many years [17].

In this study, miscanthus and switchgrass were simulated on targeted (marginal) lands identified as separate land use categories: CORM (marginal corn), COTM (marginal cotton), RICM (marginal rice), SOYM (marginal soybean), and AGRM (marginal generic agriculture). Marginal lands were identified based on two criteria: soil health issues (poorly drained and frequently flooded) and land capability classes. The total available marginal land for bioenergy crop simulation for the regular-SWAT model, developed using the 2006 land use layer, was about 8% (157 square miles) of the watershed area. Since the location-SWAT model used six different land use layers, the marginal cropland available for bioenergy crops varied over years. For the years 1992, 1999, 2001, 2004, and 2006 the available marginal land was 11% (557 square kilometers), 6% (303 square kilometers), 7% (355 square kilometers), 3% (153 square kilometers), 8% (407 square kilometers), and 10% (508 square kilometers) of the watershed area, respectively.

The SWAT crop growth model already contains parameters for a lowland cultivar of switchgrass called 'Alamo' [50]. However, two parameters: maximum potential leaf area index (BLAI) and maximum canopy height (CHTMX) were changed to represent its growth in Arkansas [41]. The value of BLAI was changed from 6 to 10 and CHTMX was changed from 2.5 m to 3 m.

Miscanthus is a relatively new crop and SWAT model lacks its growth parameters. Ng et al. [17] and Trybula et al. [51] have proposed parameter values for miscanthus modeling in SWAT. Parameter values defined by Trybula et al. [51] were used in this study to represent miscanthus in both the SWAT models. Out of 27 crop growth parameters, 22 parameters were changed, but 5 parameters were retained from the database values for switchgrass. To represent harvest scenario for miscanthus, the value of harvest efficiency (HARVEFF) was set to 0.7 and harvest index (HI) was set to 1. For more information on these parameters, please refer to the SWAT theoretical documentation and users' manual [39]. Management practices that included fertilizer and pesticides application timings and rates, tillage operations, crop planting and harvesting dates for switchgrass and miscanthus were adapted from Singh and Saraswat [41].

## 3. Results and Discussion

### 3.1. Annual Calibration

Annual calibration results are shown in Figures 2 and 3.

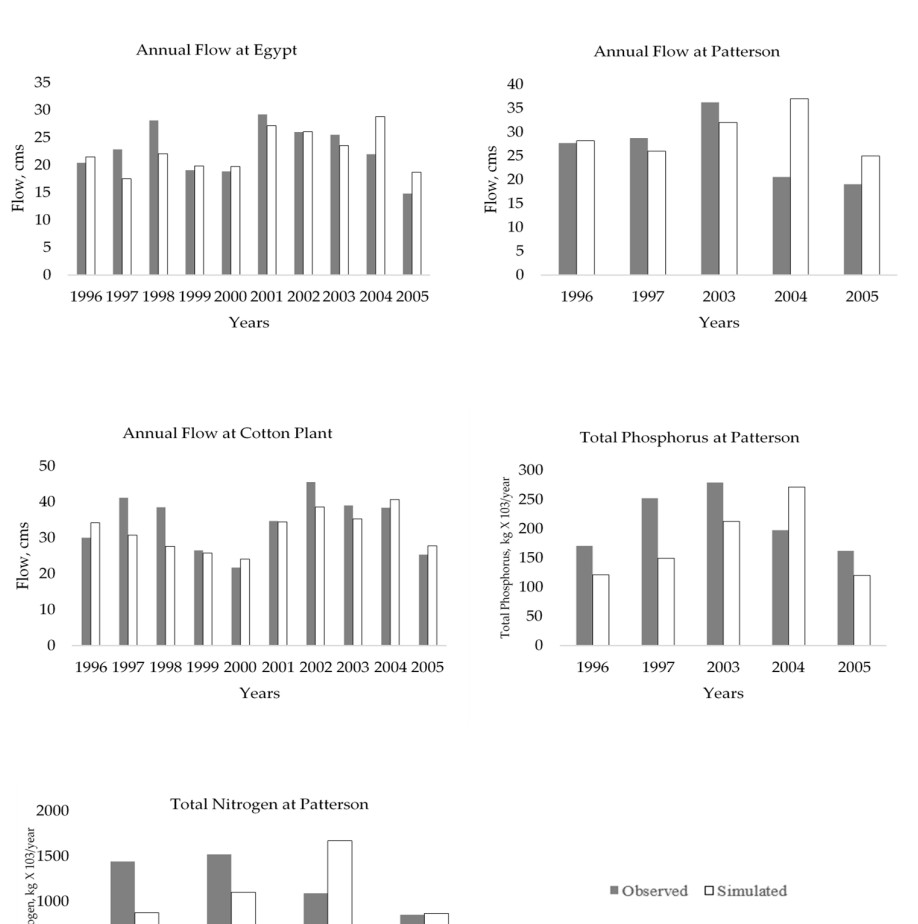

**Figure 2.** Annual calibration results for regular-Soil and Watershed Assessment Tool (SWAT) model.

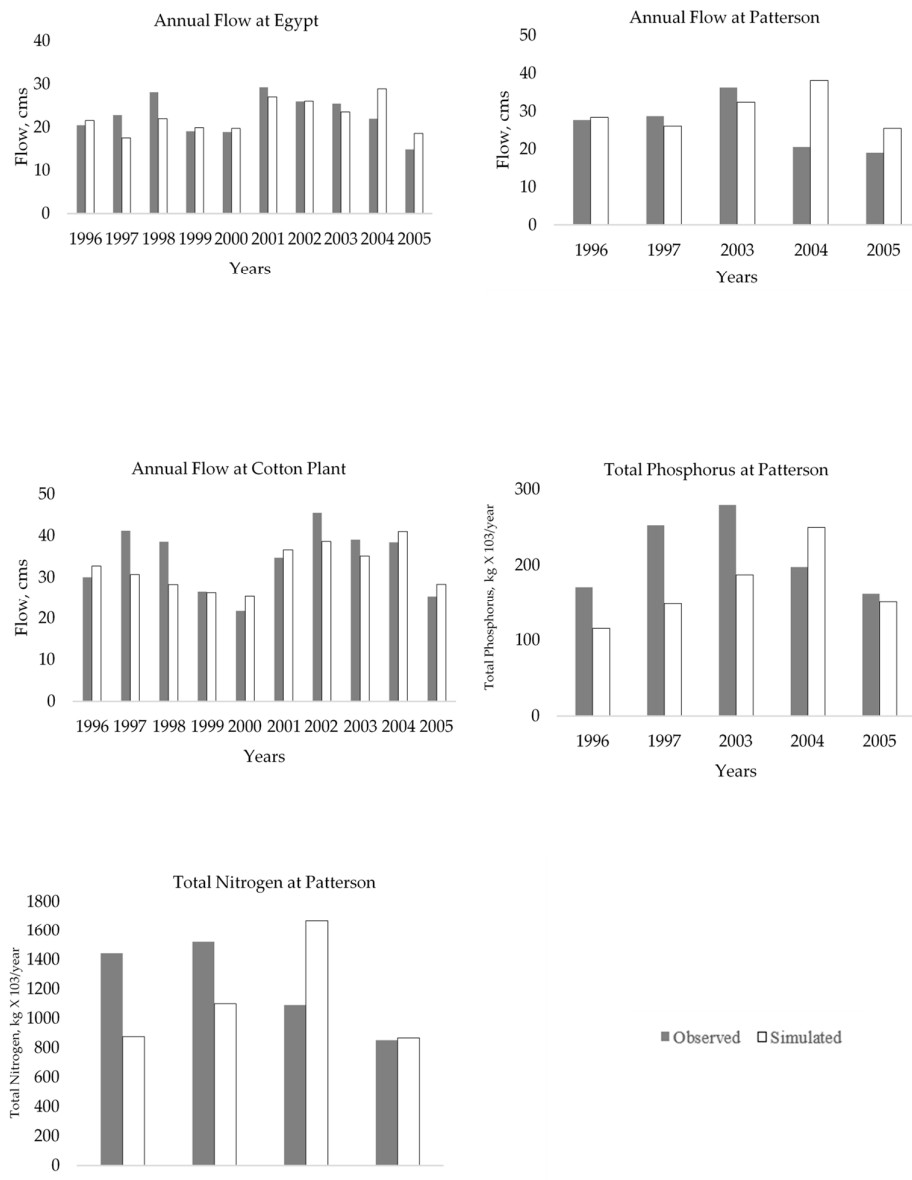

**Figure 3.** Annual calibration results for location-SWAT model.

The regular-SWAT model over-predicted total flow for 1996, 2004, and 2005 at the United States Geologic Survey (USGS)-operated gauge at Egypt; 1996, 2004, and 2005 at Patterson; and 1996, 2000, 2004, and 2005 at Cotton Plant. For rest of the years, the regular-SWAT model underestimated total flow. The location-SWAT model exhibited similar performance as it over-predicted total flow for: 1996, 1999, 2000, 2002, 2004, and 2005 at Egypt; 1996, 2004, and 2005 at Patterson USGS; and 1996, 2000, 2004, and 2005 at Cotton Plant USGS. For the rest of the years, the location-SWAT model also underestimated the flows. Past studies have indicated that the main reason for under and over prediction is spatial variability [32,47]. Against an average annual rainfall of 849.0 mm, the annual rainfall in Cache River Watershed varied from 619.7 mm (27% less) to 1353 mm (59% more), during over/under prediction period. As nutrients depend on hydrology calibration, a similar trend was seen for phosphorus and nitrogen. Both models under-predicted total phosphorus loadings during the calibration period except in 2004 at Patterson. An over-prediction in TN loads was observed in 2004

for the location-SWAT model. Both the models successfully captured hydrology (<15% RE) and nutrients (TP and TN; <25% RE) within the ranges recommended by Santhi et al. [34] except TN in the regular-SWAT model. A comparison of annual-scale observed and simulated results is presented in Tables 4 and 5.

**Table 4.** Comparison of annual-scale observed and simulated results for regular-SWAT model.

| Gauge | Output | Average | | Standard Deviation | | RE (%) |
|---|---|---|---|---|---|---|
| | | Observed | Simulated | Observed | Simulated | |
| Egypt | Total Flow (cms) | 22.7 | 21.9 | 4.5 | 3.8 | 3.6 |
| | Total Flow (cms) | 26.4 | 30.4 | 6.9 | 5.6 | 14.9 |
| Patterson | TP (kg) | 212,426 | 174,880 | 51,255.2 | 65,724.2 | 17.7 |
| | TN (kg) | 1,227,171 | 457,200 | 312,153 | 200,851.4 | 62.7 |
| Cotton Plant | Total Flow (cms) | 34.1 | 31.9 | 7.8 | 5.6 | 6.4 |

**Table 5.** Comparison of annual-scale observed and simulated results for location-SWAT model.

| Gauge | Output | Average | | Standard Deviation | | RE (%) |
|---|---|---|---|---|---|---|
| | | Observed | Simulated | Observed | Simulated | |
| Egypt | Total Flow (cms) | 22.7 | 22.5 | 4.5 | 3.8 | 0.8 |
| | Total Flow (cms) | 26.5 | 29.6 | 6.9 | 4.9 | 12.0 |
| Patterson | TP (kg) | 212,426 | 154,200 | 51,255.2 | 51,113.2 | 19.9 |
| | TN (kg) | 1,227,171 | 743,275 | 312,153 | 377,199.4 | 8.01 |
| Cotton Plant | Total Flow (cms) | 34.1 | 32.3 | 7.8 | 5.4 | 5.3 |

*3.2. Monthly Calibration and Validation*

3.2.1. Hydrology

Monthly calibration and validation results are shown in Table 6.

**Table 6.** Statistical results for regular-SWAT and location-SWAT model.

| Gauge | Monthly output | Calibration | | | Validation | | |
|---|---|---|---|---|---|---|---|
| **Regular-SWAT Model** | | $R^2$ | NSE | PBIAS | $R^2$ | NSE | PBIAS |
| Egypt | Total flow | 0.5 | 0.48 | 3.1 | 0.6 | 0.54 | 8.4 |
| | Total flow | 0.6 | 0.46 | −14.0 | 0.6 | 0.59 | −6.8 |
| Patterson | Total phosphorus | 0.6 | 0.56 | 12.3 | 0.5 | −0.41 | −7.5 |
| | Total nitrogen | 0.2 | −0.11 | 40.5 | 0.1 | −0.32 | 5.3 |
| Cotton Plant | Total flow | 0.5 | 0.55 | 6.3 | 0.6 | 0.61 | 9.0 |
| **Location-SWAT Model** | | | | | | | |
| Egypt | Total flow | 0.5 | 0.48 | 0.4 | 0.6 | 0.55 | 6.8 |
| | Total flow | 0.6 | 0.55 | −13.1 | 0.6 | 0.60 | −2.0 |
| Patterson | Total phosphorus | 0.6 | 0.57 | 30.0 | 0.5 | −0.34 | −0.1 |
| | Total nitrogen | 0.4 | −0.60 | 21.2 | 0.1 | −1.90 | 5.1 |
| Cotton Plant | Total flow | 0.6 | 0.48 | 5.4 | 0.7 | 0.59 | 9.2 |

The regular-SWAT model under-predicted total flow and surface runoff during the spring season for most of the years (1997 to 2003; 2007 and 2008; 2011 and 2012) at Egypt and Cotton Plant.

At Patterson, the model under-predicted total flow during the spring season of 1997 and 2008, but it over-predicted surface runoff in 2007 and 2009.

The location-SWAT model also under-predicted flows during the spring season for 1997, 1998, and 1999. The identical under-prediction trend was also seen for 2011 in the validation period at Cotton Plant. At Patterson, under-prediction was seen in 1997 and 2007, whereas over-prediction in flows was observed in 2009. There were multiple reasons for under- and over-prediction. Spatial variability was the primary reason affecting under- and over-predictions with watershed area being another factor affecting predictions, as the area of the CRW is relatively large. Other studies have also reported similar reasons for under and over-prediction [34,52]. In addition, high rainfall was also one of the factors affecting the flow predictions mostly from 2007 to 2010. The hydrology results were slightly on the lower side since the parameters dealing with lesser understood processes, such as subsurface flows and interaction between groundwater and rivers, became dominant in the watershed. Several parameters affecting subsurface water were sensitive for the two models which showed that subsurface flow processes were dominant in the watershed, which, in turn, added to uncertainty in models and affected calibration results [42].

### 3.2.2. Total Phosphorus

The regular-SWAT model over-predicted phosphorus loads from April to July in 2003 and January to July in 2009. Similarly, the location-SWAT model over-predicted phosphorus loads from April to July in 2003, January to July in 2009, and slightly over-predicted them from April to June in 2010. The possible reason for the over-prediction can be attributed to higher precipitation during 2003, 2009, and 2010 in the watershed. Since nutrient calibration also depends on hydrology via sediment transport, under- or over-prediction also propagated to nutrients from hydrology. During the validation period, PBIAS was in favorable range, and $R^2$ was within satisfactory range for both models, except NSE. Results reported by Bracmort et al. [53] followed a similar trend, for model performance was found satisfactory during calibration when unsatisfactory values of $R^2$ and NSE reported during validation were because of over and under-prediction of phosphorus loads. Additionally, soluble phosphorus loss can also lead to overprediction of total phosphorus, thereby impacting model performance [54]. Arabi et al. [55] also reported the unsatisfactory value of NSE during the validation period for a watershed in Indiana.

### 3.2.3. Total Nitrogen

Because calibration of nutrients depends on hydrology, a comparable trend of under-prediction was observed for TN. From March to June in 2004 and 2009, the regular-SWAT model captured the peaks. Over-predicted TN outputs were observed during June to September in 2009 and 2010, during which relatively high precipitation was received in the watershed.

The results of this study for nitrogen calibration were found reasonable when compared to other studies. Woznicki et al. [56] calibrated a SWAT model for Tuttle Creek Lake Watershed lying in Nebraska and Kansas and reported unsatisfactory results for nitrogen simulation during the calibration/validation period. The poor performance of the SWAT models was due to lack of observed data and unknown manure application rates. Glavan et al. [57] also reported unsatisfactory nitrogen simulation results because of underestimation of peak flows and a small amount of observed data. Likewise, unavailability of observed data for nitrogen and accurate planting dates posed a challenge to calibrate the models in this study. Planting dates of a crop depend on soil moisture and rainfall, so it is not possible to represent actual planting dates in crop management practices for each year for rainfall asymmetry. The planting dates, in turn, affect the nitrogen fertilizer application dates that may result in shift of peaks and affect the model calibration. It implies that there is a slight change in the crop development as per the model in contrast to the actual field performance [58].

### 3.3. Uncertainty Analysis

Uncertainties in large-scale watershed models make calibration a challenging task for process simplification, processes not accounted in the model, and processes unknown to the modeler [42]. In SUFI2, parameter uncertainty accounts for all of these uncertainties [59]. The degree to which all uncertainties are accounted for is quantified by the p-factor, which is the percentage of measured data bracketed by a 95% prediction uncertainty.

For the regular-SWAT model, 45% of observed data for flow at Egypt, 43%, 66%, and 57% of observed data for flow, TP, and TN respectively at Patterson, and 54% of observed data for flow at Cotton Plant were found to be within the 95% confidence interval of the simulation. The location-SWAT model had 61% of observed data for flow at Egypt, 67%, 69%, and 43% of observed data for flow, TP and TN respectively at Patterson, and 73% of observed data for flow at Cotton Plant within the 95% confidence interval of the best simulation.

Model uncertainties can occur due to processes that are included in a model, but their occurrences in the watershed cannot be fully understood by modelers [42]. One of the major reasons for uncertainty in flow in CRW was groundwater interactions, which could not be fully accounted for in the two models. Further, a common issue in the prediction of phosphorus due to the "second storm" effect reported by Abbaspour et al. [59] and Arnold et al. [44] was also present in both the models. The SWAT model, in general, do not have a mechanism to account for overall model uncertainty resulting from sediment and phosphorus loadings after a storm, and it overestimated the loadings instead of producing smaller loads after a similar size or a bigger storm [59]. In the case of TN, more uncertainty was observed at Patterson for the location-SWAT model. A lack of observed nitrogen data at Patterson together with dynamic land use caused shifting of peaks thereby resulting in higher uncertainty.

### 3.4. Water Quality Impacts of Bioenergy Crops

When all marginal cropland was converted in the regular-SWAT model to switchgrass, a reduction of 13.7%, 17.3%, and 15.7% was observed in sediment, TP, and TN loadings, respectively. Conversion of the marginal land to miscanthus resulted in a decrease of 13.7%, 17.2%, and 15.8% for sediment, TP loadings, and TN loadings, respectively, at the watershed outlet for the regular-SWAT model. Similarly, when all marginal cropland was converted to switchgrass, a reduction of 11.9%, 15%, and 13.6% was observed in sediment, TP, and TN loadings, respectively, and when the same land was converted to miscanthus, a reduction of 12.1%, 15.2%, and 13.3% was observed in sediment, TP, and TN loadings, respectively, at the watershed outlet for the location-SWAT model.

A decrease in total nitrogen loadings at the watershed outlet was mainly because switchgrass and miscanthus require lower nitrogen fertilizer inputs in comparison to baseline crops. Fertilizer input is a major contributor to nutrient exports from agricultural watersheds [17]. Total phosphorus loadings were observed to decrease at the watershed outlet due to absence of tillage operations after the first year of establishment of bioenergy crops. Studies have reported a decrease in sediment losses in the absence of tillage operations, which in turn affect phosphorus loadings as total phosphorus and sediments are closely related [60,61]. Moreover, a slight difference has been observed between reductions in losses for the two bioenergy crops. Crop management operations and crop growth parameters are the two factors that affect nutrient loadings. In this study, crop management operations were kept same for switchgrass and miscanthus, but crop growth parameters differed, which caused a difference in nutrient loads for the two crops.

A difference between the results for reductions in sediment and nutrient loadings for the two models was because the regular-SWAT model had a static marginal land acreage (8.2% of the total watershed area) over the entire modeling period (1992 to 2012), whereas in the location-SWAT model, the percentage of marginal lands varied during the modeling period, resulting in a change of simulated marginal acreage. The results revealed that the regular-SWAT model predicted larger sediment (1.8% more for switchgrass; 1.6% more for miscanthus), total phosphorus (2.3% more for switchgrass; 2% more for miscanthus) and total nitrogen (2.1% more for switchgrass; 2.5% more for miscanthus) losses relative to the location-SWAT. Because the location-SWAT model used six

different temporal land-use layers with varying marginal lands acreage over the modeling period, it represented the changing physical characteristics of the watershed better than the regular-SWAT. This has also been supported with the calibration and validation results. As a result, the outcomes reported in this study show that the regular-SWAT model can overpredict sediment, TP, and TN losses, thereby confounding the relative assignment of bioenergy crops on the marginal land.

The authors recommend that simulation studies evaluating impacts of bioenergy crops in other watersheds should follow use of the location-SWAT model approach for the model setup, so that the model predictions are not biased from static land-use representations. The simulated yields from both the SWAT models were found satisfactory for northeast Arkansas [58]. The regular-SWAT model simulated yields for switchgrass and miscanthus were 7.02 and 12.17 Mg/ha, respectively. Similar yield levels were simulated by the location-SWAT model for switchgrass and miscanthus, i.e., 7.55 and 11.42 Mg/ha, respectively.

Some other studies [62, 63] have also reported superior yields for miscanthus in comparison to switchgrass. Average annual yield reported for switchgrass in the US is 11.2 Mg/ha, ranging from 4.5 Mg/ha in the north to 23.0 Mg/ha in Alabama [64]. Popp [65] reported that switchgrass yields could vary from 7–12 Mg/ha on marginal lands in Arkansas. The simulated yields for switchgrass have been found to be within the recommended ranges, but more data is required to determine the best parameters to be altered in SWAT to achieve greater miscanthus yields.

## 4. Conclusions

The key conclusions from the study can be summed up as follows: (1) switchgrass and miscanthus, when grown on marginal lands, have more potential to improve water quality in the Cache River Watershed than baseline crops, (2) the location-SWAT model with dynamic land-use change is able to account for land-use change scenario in watersheds undergoing such changes, (3) the regular-SWAT model predicted larger sediment, total phosphorus, and total nitrogen loadings compared to those predicted by the location-SWAT model. Thus, the regular-SWAT showed a lower capacity to predict similar level of impact from the best management practices on water quality compared to the location-SWAT. Overall, the results of this study demonstrated that environmental impacts of bioenergy crops produced on marginal lands were better assessed using the dynamic land-use representation approach, i.e., the location-SWAT model, which would also eliminate any unintended prediction bias that gets incorporated in the model setup because of use of a static land-use layer.

**Author Contributions:** E.K. and D.S. were involved in the research design; E.K. compiled and analyzed data, setup and run the model, and drafted this manuscript under Dharmendra Saraswat's mentorship; G.S. assisted with model development and participated in manuscript preparation; E.K. has the primary responsibility for the final content; all authors have read and approved the final manuscript. All authors have read and agreed to the published version of the manuscript.

**Funding:** This research was funded by the Arkansas Natural Resources Commission vide grant number 13-600.

**Acknowledgments:** The authors would like to acknowledge the support of Arkansas High Performance Computing Center (AHPCC), University of Arkansas for providing high computing solutions.

**Conflicts of Interest:** The authors declare no conflict of interest. The funders had no role in the design of the study; in the collection, analyses, or interpretation of data; in the writing of the manuscript; or in the decision to publish the results.

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
