# Peer review of "Comparative Analysis of Bioenergy Crop Impacts on Water Quality Using Static and Dynamic Land Use Change Modeling Approach"

_water, doi:10.3390/w12020410_

Round 1

Reviewer 1 Report

In this research a widely known model was used to assess water quantity and quality in a US watershed. The view was to support that bioenergy crops improve water quality at the exit of a watershed where agricultural land dominates. Although the idea is very good and this kind of approach is important for watershed management, the manuscript could be improved before accepted for publication. Bellow are my suggestions/views for this improvement.

Abstract

It should be rewritten as the only information is the comparison between the "regular SWAT" and "location SWAT". At this point, the reader cannot know what these are and what is actually the meaning of the comparison, in terms of management and bioenergy crops production. Also, what is the point of giving the percentages of difference, are they statistically significant?

It is important to focus on policy interventions (line 14). Which are they? Not many are presented in the MS, as you actually focus on SWAT performance which is not so important for the readers.

Introduction.

An overall situation for the US is presented here. But, is SWAT the only tool/approach available? Are there any other models? Why is this chosen among others?

Materials and Methods

2.1. Some more info could be added for the soils (e.g. classification)

In addition, you use 2006 land cover data, is this due to the fact that there are not more recent data available?

2.2. In this paragraph you should focus on the way that the model works, much of this text about SWAT e.g. lines 112-126 belong to the introduction. Although the user's manual is cited, a brief description of the way that the model works is needed (via a graph or main equations). In 148 you refer that you simulate water quality, make it much more specific.

2.3. The Table 2 I think belongs to the supplementary material.

Results

lines 293-295: "exept TN in the location - SWAT" Could you discuss about this difference in a model performance and watershed management point of view? 

321-323 Precipitation favours the mobility of P? You mean via sediment transport but make it clear as P is less mobile in the soil than e.g. NO3.

A general comment is that you do not discuss your findings in terms of watershed management. Think that you have to talk about this to the stakeholders. They are not interested much about the model performance... My feeling is that there is a lot discussion on model performance and very little on what these results mean in terms of watershed management.

Please include this view or otherwise the paper does not fit to the title and the 2nd aim that you present in the introduction.

394-398 I think that the differences are not statistically important, better focus on other aspects and change also the abstract.

In the paper you do not take advantage of similar work in other (than US) watersheds. Maybe doing a small literature review and comparing to other areas it could add value to your work.

References 

The majority of them are 10 years old and little on other than SWAT related work.

Reviewer 2 Report

General comments

A useful experiment to predict water quality improvements that might be observed from planting bioenergy crops instead of food crops in marginal lands in the central USA. Modeling approach using SWAT, calibration and validation periods and statistical measures of model success are standard in the literature. Evaluation of dynamic land use capability of SWAT useful for other users of the model. Use of supercomputer to evaluate parameter sensitivity is a useful development. Writing is clear and precise. References, figures and tables are appropriate.

As detailed below, if I am reading the statistics correctly, the TN and TP results were not strong. This result needs to be more openly acknowledged and the uncertainty carried forward into discussion and conclusions. Given the difference between model outputs and observed data, how confident can we be in the bioenergy crop water quality reduction results?

Also, the monthly data should be presented in more detail. Brief summary in text could be expanded with a table or figure.

Specific Comments

Fig 1. Font too small in legend.

Line 90 Sentence should be rewritten for clarity.

Line 152. Delete ‘a’ before modified.

Table 2. Should you provide your final parameter values? Define what is meant by default value. Is this literature value or from SWAT manual?

Line 236-243. How are these results incorporated into model? Which parameters are most important to get right for model accuracy?

Line 253. Provide a definition for marginal land. Economically marginal? Why does the land fitting the description of marginal change between years?

Line 264 Rewrite sentence for clarity.

Line 288 and Line 308How do you know it is spatial variability? Is there any evidence you can point to?

Line 295 What was the RE value for TN in the location specific model?

Line 324 Should write in methods what you consider to be acceptable range for PBIAS, NSE, R2

Table 3. Extra period

Table 3. Where is the monthly data? This appears to be a summary of all the monthly data. You describe under prediction during the spring, were the other seasons more accurate? Can you provide a table summarizing seasonal data, with a measure of how it varied from year to year?

Line 324. This is problematic. An NSE of -.41 suggests poor model performance for TP during the validation phase. From my understanding NSE should be above 0.5 for good model fit? Since this is the statistic you chose to evaluate model performance, it needs more discussion. Yes R2 is reasonable and the percent bias is low, slight underprediction. But unless I am mistaken in my interpretation of NSE, you should probably be more conservative with regards to your conclusions regarding TP.

Line 322 do you mean over prediction?

Line 366 The nitrogen results do not indicate good model performance, by both R2 and NSE. PBIAS is also high during the calibration period. I think this needs to be stated more plainly. The fact that other researchers have also not had good model predictions is important to state, but I am not sure it means you can use the word “reasonable”. The small PBIAS and the large negative NSE and low R2 suggest a lot of variability in the data. I think it is important to acknowledge that and frame your results with this variability in mind. Can you give confidence intervals or standard deviations to your results in order to contrain the conclusions that are drawn? I think it is valid to say that bioenergy crops will result in reduced TN loadings, but the specific number provided is misleadingly precise.

Line 367. Is this an observation from your data or an explanatory hypothesis? Please differentiate.

Line 416. Do you mean achieve modeled miscanthus yields that are closer to observations? Unclear.

Round 2

Reviewer 1 Report

The authors have considered my recommendations and suggestions and replied adequately. The paper could be accepted for publication in Water.